# Reevaluation of Hemoparasites in the Black Spiny-Tailed Iguana (*Ctenosaura similis*) with the First Pathological and Molecular Characterizations of *Lankesterella desseri* n. sp. and Redescription of *Hepatozoon gamezi*

**DOI:** 10.3390/microorganisms11102374

**Published:** 2023-09-22

**Authors:** Yen-Chi Chang, Tai-Shen Lin, Wei-Wen Huang, Hung-Yi Lee, Cheng-Hsin Shih, Ying-Chen Wu, Chiu-Chen Huang, Ter-Hsin Chen

**Affiliations:** 1Graduate Institute of Veterinary Pathobiology, National Chung Hsing University, Taichung 40227, Taiwan; l124576930@gmail.com (Y.-C.C.); ycw0112@dragon.nchu.edu.tw (Y.-C.W.); 2Department of Post-Baccalaureate Veterinary Medicine, Asia University, Taichung 41354, Taiwanharrison0707@gmail.com (W.-W.H.); 3Department of Veterinary Medicine, National Chung Hsing University, Taichung 40227, Taiwan; 6031750317rice@gmail.com; 4Graduate Institute of Molecular and Comparative Pathobiology, National Taiwan University, Taipei 10617, Taiwan; st86123@gmail.com

**Keywords:** *Ctenosaura*, hemococcidia, hemoprotozoa, *Hepatozoon*, iguana, iguanidae, *Lankesterella*, phylogeny, refractile body

## Abstract

Hemoprotozoa are microorganisms that parasitize the blood and possess intricate life cycles. Despite the complexity of their nature, little is known about the biology of hemoprotozoa in reptilian hosts. In this study, we conducted disease surveillance on blood samples collected from six black spiny-tailed iguanas (*Ctenosaura similis*) exhibiting clinical signs. We found two different types of hemoparasites in the blood films and further confirmed they belong to the genera *Lakesterella* and *Hepatozoon* through molecular methods. In the tissue section from a dead iguana infected only with *Lakesterella* sp., parasites were also found in melanomacrophages of the liver and kidney. Since *Lakesterella* sp. infection has not been reported in *C. similis*, we propose this hemococcidian as a new species, *Lankesterella desseri* n. sp. The *Hepatozoon* parasites discovered in this study were classified as *Hepatozoon gamezi* based on their morphological characteristics, particularly the notable deformation of all infected erythrocytes, and this classification was further corroborated through molecular biological and phylogenetic analyses. This is the first hemoprotozoa investigation in *C. similis* with pathological and molecular characterization of these pathogens. We suggest that more studies are needed to understand the epidemiology, transmission, and impact of these parasites on their hosts and ecosystems.

## 1. Introduction

Hemoprotozoa constitute a group of unicellular protists that undergo diverse life cycles and develop in several tissues of both vertebrates and blood-feeding hosts [1,2]. Despite the high prevalence and diversity of hemoprotozoa in hosts of all tetrapod classes, research dedicated to understanding the interactions of these enigmatic organisms with hosts and their ecological roles remains largely unexplored, especially in herpetiles [1,3].

Haemococcidia have been described from two eimeriid families: *Lankesterella*, *Lainsonia* and *Schellackia* from the family Lankesterillidae, and *Atoxoplasma* (now considered to be Isospora species) from the family Eimeriidae [2,4]. Some authors prefer to consider *Lainsonia* as a synonym of *Schellackia* based on the morphological features. In contrast to the sexual development in the invertebrate host in other blood-borne apicomplexans, infective stages of hemococcidian do not undergo any development in the invertebrates that serve as the mechanical transmitters (paratenic hosts) [1,4]. The transmission between vertebrate hosts occurs by ingestion of infected invertebrates or through injection from the mouthpart when the vectors take blood meals [5]. To date, the definitive host species of hemococcidian are limited to birds, anurans, and lizards [2]. Due to their scarcity in domestic animal species and human beings, hemococcidians have rarely been investigated. The early literature extensively described the morphological features of hemococcidian using both light and electron microscopy [6,7,8,9]. In recent years, several studies have investigated the morphology and molecular biology of hemoparasites in wild avian species, delving even into the circadian rhythms of these parasites [3,10]. In early years, research on hemococcidia in lizards was also focused on morphological descriptions [11]. However, in recent years, with the rapid development and widespread use of molecular biology techniques and an increasing emphasis on the conservation of various species, including parasites, there have been numerous studies comprising both molecular biology screening and morphological descriptions of blood parasites in wild lizard populations [12,13,14,15,16,17]. Furthermore, some studies have also investigated the evolutionary interactions between lizard hosts and hemococcidia [18].

*Hepatozoon* parasites belong to the hemogregarines (Apicomplexa: Adeleorina), one of the most common, widely distributed, and speciose groups of reptilian hemoprotozoa [2]. *Hepatozoon* parasites are obligate heteroxenous organisms; this means that after fertilization and sporogony in various invertebrate definitive hosts that are subsequently ingested by the vertebrate host, the parasites undergo merogony in different tissues, while gamogony occurs within the blood cells. Due to the high morphological similarity in gamonts, morphological and morphometric data of gamont in the blood film cannot be the only indicators to distinguish *Hepatozoon* parasites from the other three genera in Adeleorina, not to mention the identification at the species level [1,19]. In this regard, owing to the advances in molecular technology and phylogenetic methodology, several genetic barcodes such as 18S rDNA and cytochrome c oxidase subunit I gene (COI gene) are used to provide high-resolution datasets on the phylogenetic relationships between these protozoa [2,12,17,19,20,21,22].

The black spiny-tailed iguana (*Ctenosaura similis*) is a giant iguanid lizard common in Central America. Compared with other lizards of the same genus (e.g., *C. melanosterna*), the black spiny-tailed iguana has a wider distribution area and a larger population. It is listed as a least-concern species on the IUCN Red List [23]. However, there are few studies or records describing the infectious organisms carried by black spiny-tailed iguanas, especially protozoa. Recently, we had the opportunity to investigate the blood parasites of the black spiny-tailed iguana population to contribute more advanced knowledge of the morphological, pathological, and molecular phylogenetic aspects of this neglected host and its parasites.

## 2. Materials and Methods

### 2.1. Animals, Clinical Examination, and Sampling

In October 2021, 6 juvenile black spiny-tailed iguanas (*Ctenosaura similis*) were imported from Nicaragua to Taiwan after being wild-caught. These iguanas were distributed to two different owners in the city of Taichung. These iguanas were collectively or individually housed in reptile terrarium tanks measuring 60 × 60 × 45 cm. These iguanas were not co-housed with other reptiles. The basking spot within the tank maintained a temperature of approximately 30 degrees Celsius. The substrate used was red clay soil. The lizards’ diet included vegetables, crickets, and superworms. According to both owners, all iguanas developed clinical signs including stunt growth, depression, anorexia, ectoparasite infestation, and sudden death. In March 2022, an iguana (case 1) with a clinical history of thermal burn was found dead. The carcass was admitted to the clinical microbiological laboratory of National Chung Hsing University by Owner 1 for pathological examination.

Thereafter, we contacted Owner 2 to confirm the clinical status of the iguanas. Owner 2 also mentioned the symptoms of ectoparasite infestation and stunted growth in these iguanas. In June 2022, we conducted clinical observation and sampling on the remaining five surviving black spiny-tailed iguanas (cases 2 to 6). The body weight of each iguana was measured and photos were taken to record the infestation status of ectoparasites and other anomalies including trauma, dyskeratosis, and emaciation. Blood samples from each iguana were collected using a syringe with a 25-gauge needle through venipuncture of the ventral coccygeal vein for subsequent clinical pathological and molecular examinations [24]. After preparing blood films, the fresh blood samples were stored in heparin and EDTA anticoagulated tubes. The ectoparasites were brushed down and collected in sterilized Eppendorf tubes. Swabs of the oral cavity and cloaca were also collected. The heart blood of Iguana 1 was obtained during necropsy for blood film preparation. However, due to the insufficient volume of the blood sample and post-mortem clotting, hematological and serum biochemistry examinations were not performed in iguana 1.

### 2.2. Clinical Pathological Examination

Blood smears were air-dried and fixed for 5 min in absolute methanol. All blood smears were stained with Diff-Quik (sysmex, Hyogo, Japan) according to the manufacturer’s instructions. Slides were examined under the BX51 light microscope (Olympus, Tokyo, Japan) equipped with a DP21 digital microscopy camera (Olympus, Japan) for hemoparasites counting to 15,000 erythrocytes [25]. The microphotographs of the sporozoites were captured using a digital camera and the length and width of the sporozoites were measured using the free software MB-ruler 5.0 (http://www.markus-bader.de/MB-Ruler/index.php, accessed on 8 August 2023) [4]. The heparinized serum samples of iguanas 2 to 6 were used for serum biochemistry on the VetScan VS2 Analyzers with Avian/Reptilian Profile Plus reagent rotors (Zoetis, Parsippany-Troy Hills, NJ, USA).

### 2.3. Pathological Examination

During the necropsy of iguana 1, the major organs including the heart, liver, spleen, lungs, kidney, brain, gastrointestinal tract, urine bladder, thigh muscle, vertebrate, gonads, thyroid gland, adrenal glands, and trachea were examined, photographed, collected, and fixed in 10% neutral-buffered formalin for histopathology. Frozen tissue samples of the lungs and liver were stored at −20 °C until use in nucleic acid extraction. Fixed tissue samples were trimmed, embedded in paraffin, sectioned at 5 µm, and stained with hematoxylin and eosin for routine histopathological examination. Briefly, all slides were examined under a BX51 light microscope (Olympus, Tokyo, Japan) equipped with a DP21 digital microscopy camera (Olympus, Tokyo, Japan). Histopathological changes including edema, hemorrhage, leukocyte infiltration, degeneration, necrosis, and the presence of microorganisms were recorded and captured using the digital camera.

### 2.4. Nucleic Acid Extraction

The anticoagulated blood samples were directly used in DNA extraction, while the liver sample of iguana 1 and mites collected from each lizard were individually homogenized using the Biomasher II homogenizer (Nippi, Tokyo, Japan) and the provided grinding microcentrifuge tubes prior to use in DNA extraction. DNA extraction was performed using the Tissue/Blood DNA Mini Kit according to the manufacturer’s instructions (Geneaid, New Taipei City, Taiwan).

### 2.5. Polymerase Chain Reaction—Partial 18S rDNA and Partial mt COI of the Apicomplexan Parasites

PCR reactions to amplify the partial 18S ribosomal DNA of apicomplexan parasites were performed using the primer set BT-F1/EimIsoR1 or BT-F1/Hep1600R [4]. Partial mitochondrial cytochrome c oxidase subunit I (COI) DNA was obtained by using the Cocci_COI_For/Cooci_COI_Rev primer set [26]. PCR reactions (total volume of 25 μL) composed of 2 μL of the DNA template, Taq DNA Polymerase Master Mix RED (Ampliqon, Odense, Denmark), and 200 nM of each primer were used. Using a MiniAmp Plus thermal cycler (Thermo Fisher Scientific, Waltham, MA, USA), reactions were run using the following conditions: 95 °C for 3 min, 35 cycles at 95 °C for 30 s, annealing temperature at 58 °C for 45 s, 72 °C for 105 s, and a final extension at 72 °C for 7 min. The sample obtained from a confirmed case of *Isospora amphiboluri* infection in a bearded dragon was used as positive control and nuclease-free water (Qiagen, Hilden, Germany) was used as a negative control in parallel in each PCR run. PCR reaction products were electrophoresed using a 1.5% submarine agarose gel, stained with ethidium bromide and visualized using UV trans-illumination. All amplicons were recovered from agarose gels with surgical scalpels and subjected to Sanger’s sequencing (Tri-I Biotech Inc., New Taipei City, Taiwan). The amplicons whose chromatograms revealed multiple peaks were cloned by using the TOPO^®^ TA Cloning^®^ Kit with One Shot^®^ TOP10 Electrocomp™ *E*. *coli* (Thermo Fisher Scientific, USA). PCR screening was performed in all colonies with the same PCR condition. In the positive colonies, the plasmids were purified using the Axygen^®^ AxyPrep™ Plasmid Miniprep Kit (Axygen Biosciences, Union City, CA, USA) according to the manufacturer’s instructions.

### 2.6. Immunohistochemistry

The tissue sections of iguana 1 were placed on silane-coated slides and heated overnight in an oven at 56 °C, after which they were submitted to immunohistochemistry (IHC) (BOND BIOTECH Inc., Taichung City, Taiwan). The application of a rabbit polyclonal antibody against *Toxoplasma gondii* was performed to validate the cross-immunoreactivity, which was reported in a previous study of *Lankesterella* sp. infection in White’s tree frogs [27]. 

### 2.7. Phylogenetic Analysis

After Sanger’s sequencing, the forward and reverse sequences were assembled and optimized using DNASTAR Lasergene (DNASTAR^®^) SeqMan software 7.1.0.4 [28]. Each new 18S sDNA/COI sequence was approximately 1600 and 800 bp in length, respectively. Newly generated and existing sequences from GenBank were aligned using the MUSCLE algorithm with default settings in MEGA X [29]. Nucleotide evolutionary models were evaluated using the model selection software package in MEGA X [29]. The details of alignment are listed in Section 2.7.1 and Section 2.7.2. The best-fitting substitution model, according to the Akaike Information Criterion evaluation of hierarchical likelihood ratio tests, suggested the general time reversible model with rates that vary over sites according to the invariable sites plus gamma distribution (GTR+G+I) for all sequence datasets in MEGA X [29]. For each dataset, the maximum likelihood method (ML) and the Bayesian inference method were conducted using MEGA X and MrBayes version 3.2.6, respectively [29,30]. The ML bootstrap method with 1000 replicates was used to assess the confidence of the trees. Gaps were treated as missing in all datasets. Bayesian analysis consisted of 2 runs of 4 chains each with 2,000,000 generations per run and a sampling interval of every 200 generations. A “burn-in” of 500,000 generations was applied and 30,000 trees were obtained for the consensus trees. Support values are labeled by the nodes as either bootstrap value or posterior probability. The consensus trees were visualized in Interactive Tree Of Life v5 [31]. In addition, the built-in function in MEGA X was used to calculate the pairwise distances between sequences.

#### 2.7.1. Phylogenetic Analysis of *Lankesterella* n. sp.

The final alignment of 18S rDNA sequences contained 1394 positions and 132 sequences composed of 18 newly obtained *Lankesterella* sequences and 114 sequences retrieved from the GenBank database. The sequences belonging to the family Sarcocystidae (*Cystoisospora* sp., *Toxoplasma* sp., *Sarcocystis* sp., *Neospora* sp., and *Frenkelia* sp.) were used as outgroups. In contrast to 18S rDNA sequences, the registered COI sequences in the GenBank were relatively scarce and sequences of several species were not available (e.g., *Schellackia* sp.). The final alignment of COI sequences contained 747 positions and 90 sequences composed of 1 newly obtained *Lankesterella* n. sp. sequence from iguana 1 and 89 sequences retrieved from GenBank. Family Sarcocystidae sequences served as outgroups (*Cystoisospora* sp., *Toxoplasma* sp., and *Neospora* sp.). The GenBank accession numbers of sequences used in the phylogenetic analysis are given in Appendix A.

#### 2.7.2. Phylogenetic Analysis of *Hepatozoon gamezi*

The final alignment of 18S rDNA sequences contained 1605 positions and 152 strains composed of 14 newly obtained *Hepatozoon* sequences and 138 sequences retrieved from the GenBank database. The sequences of *Dactylosoma* sp., *Adelina* sp., and *Haemogregarina* sp. were used as outgroups. The GenBank accession numbers of sequences used in the phylogenetic analysis are given in Appendix A.

## 3. Results

### 3.1. Clinical Examination

#### 3.1.1. Physical Examination, Hematology, and Serum Biochemistry

In iguana 2–6, a total of 5 lizards, numerous external parasites (mites) measuring approximately 0.2 to 1.0 mm in size, ranging from orange-red to dark red in color, were observed bearing between scales (Figure 1). Table 1 shows the results of the physical examinations and the biochemical profiles of 6 iguanas. In iguana 1, only hydration status was conducted. All iguanas were stunted and dehydrated. Given the absence of reference values of serum biochemistry items of any *Ctenosaura* sp., the parameter values were based on tables of juvenile green iguanas listed in *Exotic Animal Formulary, 5th Edition* [32]. The body condition scores (BCS) were higher in iguanas with lower parasitemia of hemococcidia (iguanas 1 and 4) than those iguanas with higher parasitemia of hemococcidia. Except for values of albumin, globulin, and total protein being higher than the reference parameters of the green iguana, the serum biochemical profile did not reveal any obvious abnormalities. Notably, the values of globulin were higher in iguanas with higher parasitemia of hemococcidia.

#### 3.1.2. Morphological Characterization of Hemococcidia

We observed sporozoites infecting erythrocytes in five of six thin blood smears of the *C. similis*. No sporozoite of hemococcidian was observed in the blood smear of iguana 5. The mean intensity per 15,000 erythrocytes in the five positive smears was 18.6. The highest intensity was observed in iguana 4 with 48/15,000 erythrocytes, and the lowest intensity was observed in iguana 2 with 6/15,000 erythrocytes. The mean size of the sporozoites was 12.25 × 3.71 μm (Table 2), and all sporozoites were identified in erythrocytes (Figure 2) except for a sporozoite in one monocyte of iguana 1 (see Section 3.2). The parasites were elongated, with one pointed end and opposite rounded ends (Figure 2a,c,d). Occasionally, the sporozoites bent and formed circular, lentiform, or teardrop-shaped structures with the two ends adjacent to each other (Figure 2b,d). Displacement of the nuclei of host cells was mild. The cytoplasm remnants were well-visible in infected cells. The sporozoite cytoplasm stained moderately amphophilic and the outline was even. The nucleus of the sporozoite was usually located closer to the pointed end and appear either adhered to one side of the sporozoite or in broad-band form that occupied the entire width of the parasites.. In all sporozoites examined in this study, one to two bluish, round to oval refractile bodies, each measuring approximately 1.4 to 3.1 μm in diameter, could be observed either posterior to the sporozoite nucleus or both anterior and posterior to the nucleus (arrows in Figure 2).

#### 3.1.3. Morphological Characterization of *Hepatozoon* sp.

We observed gamonts infecting erythrocytes in five of six thin blood smears of the *C. similis*, except for iguana 1. The mean intensity per 15,000 erythrocytes in the five positive smears was 8.8. The highest intensity was observed in iguana 3 with 18/15,000 erythrocytes, and the lowest intensity was observed in iguana 2 with 2/15,000 erythrocytes. The gamonts were rod-shaped with centrally located nuclei. In only one erythrocyte of iguana 5, a slender and elongate immature gamont was noted, measuring 12.3 × 3.2 μm in size, with eosinophilic cytoplasm (Figure 3a). The advanced gamonts were separated into two groups based on the morphology of the cytoplasm: (1) vacuolated form: translucent to pale eosinophilic cytoplasm with vacuoles varying in size; (2) basophilic form: strong basophilic cytoplasm with finely eosinophilic granules (Figure 3). The mean sizes of the gamonts were 16.55 × 7.61 and 17.00 × 7.54 μm, respectively. All gamonts were identified in erythrocytes. In all infected erythrocytes, the nucleus was displaced eccentrically by the gamonts, but no obvious deformation was observed. In addition, the cytoplasm of all infected erythrocytes became pale and translucent, and fusiform erythrocytes characterized by bilateral tapered extension were often noted (white arrow of Figure 3c inset). Occasionally, dual infection of hemococcidan sporozoite and *Hepatozoon* gamont can be observed in one erythrocyte (Figure 3d). 

### 3.2. Postmortem Examination in Iguana 1

Iguana 1 was dehydrated and emaciated, with muscle loss on extremities and tail base. The ventral surface of the body, specifically the anterior one-third, showed extensive desquamation of the skin, exposing underlying tissues. Additionally, a significant amount of yellow-brown fibrinous exudate was observed adhering to the exposed areas. At necropsy, the fat pads were atrophic and no visceral fat was observed. The liver was diffusely dark green and slightly swollen withround edges (Figure 4). The serosa of the kidneys also revealed several dark green flat foci 0.1–1.0 mm in diameter. A heart blood smear was performed. The parasites consistent with hemococcidian sporozoites, as mentioned in Section 3.1.2, could be observed in erythrocytes and in one monocyte (Figure 5). Under histopathological examination, the melanomacrophage centers were markedly hyperplastic (Figure 6). Within the cytoplasm of the melanomacrophages, protozoa were noted individually or in clusters, being oval to fusiform in shape, approximately 9.0 × 1.7 μm in size, with a centrally located nucleus (arrow in inset of Figure 6). A focal area of the subserosa of the kidney also revealed the aggregation of melanomacrophages with the same intracytoplasmic protozoa. 

Other pathological changes, including locally extensive fibrinonecrotic dermatitis of the ventral side that was most likely due to thermal burn, follicular degeneration of ovaries, and mild gastritis with a low number of ascarid nematodes infection, were all thought to be independent of the hemococcidian infection. No evidence of *H. gamezi* infection was noted in all organs of iguana 1.

### 3.3. Immunohistochemistry

The liver section of iguana 1 was negative against the anti-*Toxoplasma gondii* antibody.

### 3.4. Polymerase Chain Reaction and Nucleotide Sequence Analysis

All iguana samples were positive for the partial 18S rDNA PCR, while only mites from iguana 6 were positive. The amplicon of iguana 1 revealed no multiple peaks in the chromatogram. In contrast, TA cloning had been performed on PCR amplicons from iguanas 2 to 6 due to the presence of multiple peaks in chromatograms, and all colonies were submitted for bilateral sequencing. Homology searches of nucleotides were performed using the Basic Local Alignment Search Tool (BLAST) algorithm at the National Center for Biotechnology Information (NCBI). Surprisingly, both *Lankesterella* and *Hepatozoon* sequences were obtained from colonies of iguana 2, 3, 4, and 6. In iguana 5, only four haplotypes of *Hepatozoon* sequences were obtained. The new sequence data of partial 18S rDNA of *Lankesterella* and *Hepatozoon* are available in the GenBank database under the accession numbers OR425015–OR425032 and OR425034–OR425047, respectively.

In contrast with the high detection rate of the partial 18S rDNA PCR, COI PCR yielded positive results only in the liver sample of iguana 1. Homology searches using the BLAST algorithm revealed that the amplified sequence showed high homology with sequences of *Lankesterella* parasites. The new sequence data of COI are available in the GenBank database under the accession number OR427298.

### 3.5. Phylogenetic Analysis

#### 3.5.1. Phylogenetic Analysis of *Lankesterella desseri* n. sp.

Both Bayesian inference and maximum likelihood methods yielded 18S rDNA phylogenetic trees with nearly identical topology. The newly identified haplotypes collectively formed an extensive monophyletic clade, closely aligned with sequences of *Lankesterella* parasites reported in earlier studies (Figure 7). Among the 18 new haplotypes from the black spiny-tailed iguanas, 14 of them were clustered with haplotypes DD1 and DD4 (accession numbers MF167547 and MF167548) obtained from the desert iguana *Dipsosaurus dorsalis*, showing relatively strong support (bootstrap value = 66). The other four haplotypes from iguanas 3 and 6 form a monophyletic clade without other sequences (bootstrap value = 100). The sister groups of this new clade include DD2 and DD3 from the same desert iguana *D. dorsalis* of haplotypes DD1 and DD4 and sequences of *Lankesterella* parasites detected in various areas and host species (reptiles, amphibians, and birds). As reported in previous studies, the sequences of *Schellackia* parasites formed a monophyletic clade, while no sequences of *Lankesterella* parasite were included [4,12].

In a phylogenetic analysis of the partial COI gene, the newly obtained sequence in this study formed a well-supported clade (posterior probability = 100) with the closest sister group to the four available COI sequences of *Lankesterella* parasites in the GenBank database (Figure 8). The sister group of the *Lankesterella* clade consisted of different genera in the Eimeriorina, including *Isospora*, *Caryospora*, and *Eimeria* from reptiles, marsupials, and house shrews (*Suncus* sp.).

#### 3.5.2. Phylogenetic Analysis of *Hepatozoon gamezi*

The 18S rDNA phylogenetic trees generated by the Bayesian inference method and the maximum likelihood method shared nearly identical topologies (Figure 9). All *Hepatozoon* sequences obtained in this study formed a new monophyletic clade with high statistical support (bootstrap value = 87 and posterior probability = 100) and were inserted into a large clade composed of *Hepatozoon* strains obtained from reptilians or rodents. Nevertheless, two *Hepatozoon* haplotypes of iguana 3 (B3-1 and B3-4) and one haplotype of iguana 6 (B6-9) were separated from other haplotypes with high nodal supports (posterior probabilities = 100).

### 3.6. Description of New Lankesterella Species and Redescription of Hepatozoon gamezi

#### 3.6.1. *Lankesterella desseri* n. sp.

Type host: The black spiny-tailed iguana *Ctenosaura similis* Gray, 1831 (Iguanidae, Iguania, Sauria, Squamata).

Additional hosts: Unknown.

Type locality: Nicaragua.

Site of infection: Erythrocytes, rarely observed in monocytes.

Prevalence: 83.3% (five out of six examined *C. similis* were infected).

Type specimens: Hapantotype (thin blood film and histological section, collection number CO22-02023, juvenile *Ctenosaura similis* with parasitemia of 6/15,000 erythrocyte, Diff-Quik stained, National Chung Hsing University, collected 26 June 2022 by Y.C. Chang). A co-infection with *Hepatozoon gamezi* is present in the type material.

DNA sequences: 18S ribosomal DNA gene lineage L1 (1578 bp, GenBank accession number OR425017).

Distribution: This infection has been reported only in the type hosts. There is no sequence of 100% similarity deposited in GenBank.

Etymology: The parasite is named in honor of Dr. Sherwin S. Desser, Professor Emeritus of the University of Toronto, in recognition of his contribution to the investigation and exploration of protozoa in wildlife species, especially the detailed description of hemoparasites in *C*. *similis* [11].

Blood stage (Figure 2 and Figure 5): The sporozoites develop mostly in erythrocytes, and the range of size is provided in Table 2. These parasites are fusiform and smooth in outline, with one pointed end and the opposite rounded ends. The cytoplasm of sporozoite shows an eosinophilic and granular appearance. One a bluish-grey color, spherical, prominent inclusion posterior to the nucleus of the sporozoite, in a proportion of sporozoites with another smaller inclusion anterior to the nucleus, can be observed (refractile bodies, RB). The nuclei of sporozoites are in broad-band form and usually occupy the entire width of the sporozoites. The parasites usually abut against the nuclei of the host cells, but the displacement or distortion of both nuclei and cytoplasm are mild. In some bent sporozoites, protruding of the cell membrane of host cells due to the compression of parasites can be seen. 

Liver stage (Figure 6): Under H&E stain section, the melanomacrophage center (MMC) was hyperplasic. Within the cytoplasm of the melanomacrophages, protozoa were noted individually or in clusters, oval to fusiform in shape, approximately 5.0 × 1.5 μm in size, with centrally located nuclei.

Taxonomic remarks: The presence of RB is thought to be a distinctive morphological feature of the sporozoites in the hemococcidian genera *Lankesterella* and *Schellackia* [4,5,6,33]. Hence, *Lankesterella desseri* n. sp. can be distinguished from other hemoparasites with similar morphological features, such as parasites in *Hepatozoon* and *Hemogregarine* genera. To the best of our knowledge, there were no decisive morphological characteristics of sporozoite to distinguish the parasites of the genus *Lankesterella* from the parasites of the genus *Schellackia*. To date, the major differences between genera *Lankesterella* and *Schellackia* are the geographical region of host species and the different evolution patterns in molecular phylogeny [2,4,12]. A search of the previous literature revealed similar descriptions in two publications. Dr. Desser had described *Schellackia* sporozoites infection in three *C. similis* in Costa Rica [11]. The sporozoites share common features with the parasites we found in this study, including the size of sporozoites and the presence of refractile bodies. Davis et al. described *Hepatozoon* sp. infection in the black-chested, spiny-tailed iguana (*C. melanosterna*) [34]. However, they were unsuccessful in amplifying the 18S rDNA fragment using the universal *Hepatozoon* primer set. In addition to the morphological and morphometric similarity between *Lankesterella desseri* n. sp. and the *Hepatozoon* sp. mentioned in their research, the oval-shaped variant of parasites described by the authors aligns with the lentiform sporozoites of *Lankesterella desseri* n. sp. observed in our study. Nonetheless, molecular data could not be obtained from these two publications, thereby hindering the further establishment of a demonstrable relationship between these parasites.

#### 3.6.2. *Hepatozoon gamezi*

Type host: The black spiny-tailed iguana *Ctenosaura similis* Gray, 1831 (Iguanidae, Iguania, Sauria, Squamata).

Additional hosts: Unknown.

Type locality: Nicaragua.

Site of infection: Erythrocytes.

Prevalence: 83.3% (five out of six examined *C*. *similis* were infected).

Type specimens: Hapantotype (blood film collection number B3, juvenile *Ctenosaura similis* with parasitemia of 13/15,000 erythrocyte, Diff-Quik stained, National Chung Hsing University, collected 26 June 2022 by Y.C. Chang). A co-infection with *Lankesterella desseri* n. sp. is present in the type material.

DNA sequences: 18S ribosomal D Nicaragua.NA gene lineage B3-14 (1602 bp, GenBank accession number OR425037).

Distribution: This infection has been reported only in the type hosts. There is no sequence of 100% similarity deposited in GenBank.

Vector: Unknown.

Etymology: The parasite is named in honor of Dr. Rodrigo Gamez Lobo, the Director of Costa Rica’s Instituto Nacional de Biodiversidad (INBIO), in recognition of his contribution to the conservation of lizard hosts and protozoan parasites in Costa Rica’s wildland [11].

The gamonts (Figure 3) develop in erythrocytes and the range of size is provided in Table 2. These parasites are rod-shaped with centrally located nuclei. The cytoplasm of the gamonts can be either pale eosinophilic with vacuoles varying in size or strong basophilic with finely eosinophilic granules. In all infected erythrocytes, the nucleus was displaced eccentrically by the gamonts but no deformation was observed. In addition, the cytoplasm of all infected erythrocytes became pale and translucent, and fusiform erythrocytes characterized by bilateral tapered extension were often observed.

Taxonomic remarks: The morphological and morphometric characteristics of the gamonts and the cytopathological effect are identical to the description in the previous publication [11]. As a result, we highly suggest that the parasites observed in our study are *Hepatozoon gamezi*. The molecular data and the result of phylogenetic analysis are consistent with the ultrastructural findings in the publication of Dr. Sherwin S. Desser, indicating that these hemoparasites belong to the *Hepatozoon* genus [11].

## 4. Discussion

In this study, we successfully identified *Hepatozoon gamezi* and a new species, *Lankesterella desseri* n. sp., in diseased black spiny-tailed iguanas using a combination of clinical pathology, molecular biology, and histopathology methods. We conducted morphological descriptions and molecular phylogenetic analyses on these two blood parasites. To the best of our knowledge, this is the first study analyzing sequence data of hemococcidia and *Hepatozoon gamezi* from the black spiny-tailed iguana.

The literature that mentioned *Lankesterella* species predominantly consists of infection case reports and phylogenetic analyses in wild birds and anurans [3,5,7,7,10,27,35]. In contrast, emphasis on *Lankesterella* parasites that infect reptilians has been limited [4,13]. In our research findings, hemococcidia sporozoites were observed in both blood smears and tissue sections. Furthermore, under histopathological examination, we observed that these sporozoites activated innate immunity (MMCs) [36]. The morphological descriptions align with previously discovered hemococcidia in anurans and several lizard species [9,11,27,35,37,38,39]. Subsequently, we used molecular biology techniques to further classify the hemococcidia we identified at the genus level. In a previous study conducted by Megía-Palma et al. in 2017, the association between morphology and molecular phylogenetics in hemoparasites was investigated in three lizard species from North America and two from South America [4]. The results revealed that the molecular phylogenetic classification of hemoparasites is related to the geographical pattern rather than the morphology of these hemococcidia. In our study, the sporozoites with refractile bodies, a distinctive morphological characteristic compatible with the sporozoites in the hemococcidian genera *Lankesterella* and *Schellackia*, were observed in the blood film of *C. similis*. Molecular phylogenetic results demonstrate that the newly obtained sequences are grouped within clades containing other *Lankesterella* species, corroborating previous research indicating that New World lizards are infected by *Lankesterella* rather than *Schellackia* parasites [4]. In addition, the obtained sequences are segregated into two distinct clades: (1) the desert iguana (*Dipsosaurus dorsalis*) forms a clade with DD1 and DD4, and (2) forms a monophyletic clade without other sequences. The sister groups of this new clade include DD2 and DD3 from the same desert iguana of haplotypes DD1 and DD4, and sequences of *Lankesterella* parasites detected in various areas and host species (reptiles, anurans, and birds). The haplotype diversity of *Lankesterella desseri* is in line with previous studies that mentioned the haplotype abundance in different hosts [4,10,18]. The high resemblance in the evolutionary process of *Lankesterella* parasites from *C. similis* and *Dipsosaurus dorsalis* may be explained by the combined influence of the close phylogenetic relationship between these two lizard species and the demonstrated host–parasite cospeciation observed in *Schellackia* parasites infecting the family Lacertidae [18]. Other possible causes include habitat overlapping of the lizard hosts and parallel evolution of these hemococcidia. Additional biological information and further investigation are necessary to fully understand the evolutionary interaction between the black spiny-tailed iguana population and hemococcida parasites. In the phylogenetic tree of the COI gene, the *Lankesterella* branches form a distinct clade, indicating the discriminative potential of the COI gene at the species level [26]. However, this clade is closely related to several different genera within Eimeriorina from reptiles, marsupials, and house shrews. The possible reasons for such results conflicting with traditional classification could be attributed to the insufficiency of the COI gene database or the true conflict between molecular phylogenetics classification and traditional morphological classification in eimeriids, as previously mentioned in several studies [40,41]. With no prior reports of black spiny-tailed iguanas infected with *Lankesterella* sp. and forming an independent clade in molecular phylogeny, we describe a new parasite species, *Lankesterella desseri* n. sp.

In addition to hemococcidians, the blood smears of the black spiny-tailed iguanas also contain large, short rod-shaped, variably staining structures, closely resembling the morphology of *Hepatozoon gamezi* that was identified in black spiny-tailed iguanas by Desser et al. in 1997 [11]. The molecular biological examination of our study also corresponds to the ultrastructural findings in the previous study. Hence, we suggest that the parasites are *Hepatozoon gamezi*. The phylogenetic analysis of 18S rDNA revealed that all *H. gamezi* form a distinct clade. To our knowledge, this is the first time that the nucleotide sequence of *H. gamezi* has been obtained and their phylogenetic analysis has been performed. Morphologically, infected red blood cells exhibited significant deformations and faint staining. Although the present study did not find *H. gamezi* in the tissue section of iguana 1, the potential impact on the host as described in other *Hepatozoon* sp. cannot be disregarded [42,43]. We conducted PCR screening targeting hemoprotozoa on mites from each iguana, and *Hepatozoon gamezi* was detected in the mites parasitizing iguana 6. However, due to the inability to identify any life stage of *Hepatozoon* parasites within the tissue section of mites, and the possibility that the positive PCR signal may be a result of mites feeding on the iguana blood, we suggest that further information is required to elucidate the role that mites play in the life history of *Hepatozoon gamezi* (e.g., whether the mite act as final hosts or mechanical vectors).

In this study, we identified two parasites, *L. desseri* n. sp. and *H. gamezi*, from clinically diseased black spiny-tailed iguanas. We conducted morphological and molecular phylogenetic analyses of these parasites, along with clinical and histopathological examinations revealing notable pathological changes induced by these blood parasites in the host lizards. Considering their pathogenicity and potential for interspecies transmission to other lizards or even native reptiles, we propose the need for comprehensive blood parasite and ectoparasite surveys of related lizard species, alongside the implementation of relevant quarantine measures in the growing exotic pet trade.

## Figures and Tables

**Figure 1 microorganisms-11-02374-f001:**
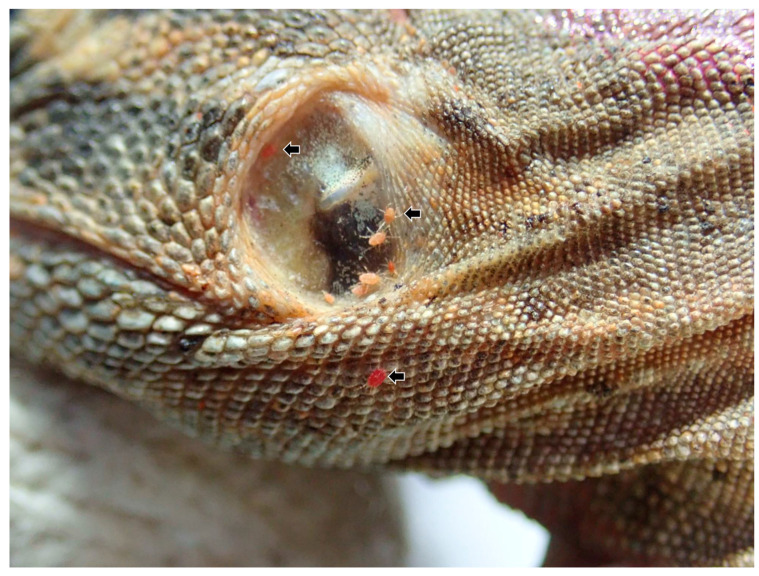
Mite infestation of a black spiny-tailed iguana, iguana 2. The iguana is dehydrated, with numerous mites surrounding the tympanic membrane (arrows).

**Figure 2 microorganisms-11-02374-f002:**
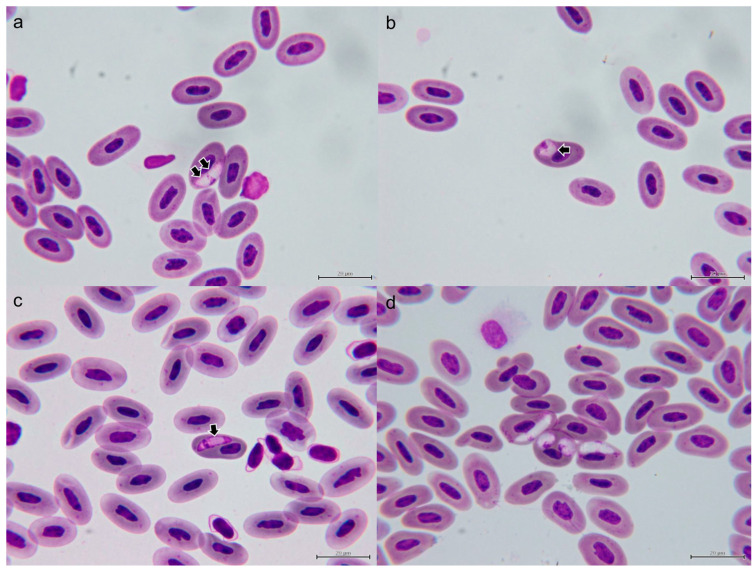
Sporozoites of *Lankesterella desseri* n. sp. infected the erythrocyte of iguanas 3 (**a**,**b**), 6 (**c**), and 5 (**d**). One to two bluish, round to oval refractile bodies can be observed either posterior to the sporozoite nucleus or both anterior and posterior to the nucleus (arrows). Sporozoites bent in circular or lentiform can be observed (**b**,**d**). Infrequently, two sporozoites infecting one erythrocyte can be seen (**d**). Deformation of the erythrocytes is absent (**a**) to mild (**b**–**d**).

**Figure 3 microorganisms-11-02374-f003:**
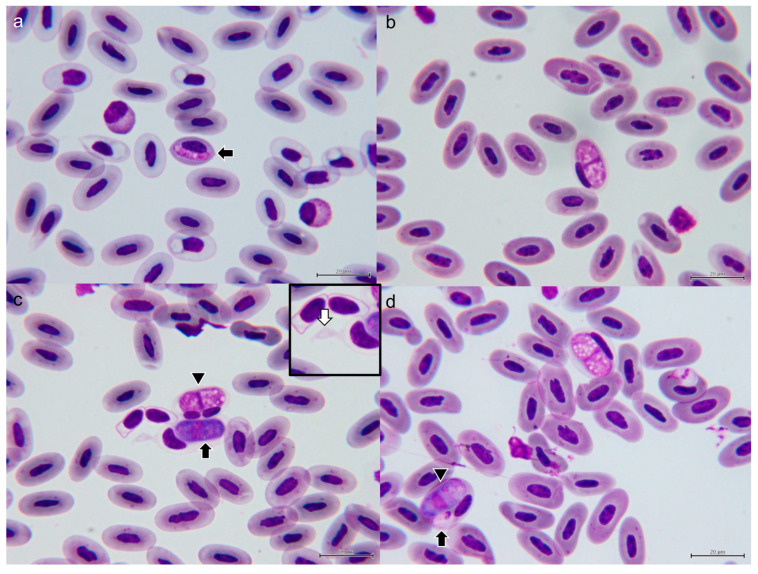
Gamont of *Hepatozoon gamezi* infected the erythrocyte of iguanas 5 (**a**), 3 (**b**), 6 (**c**), and 5 (**d**). Immature gamont is smaller and thinner than mature gamont (arrow in (**a**)). Mature gamont presents in either vacuolated, eosinophilic form ((**b**) and arrowhead in (**c**)) or basophilic form (black arrow in (**c**)). Bilateral tapered extension of infected erythrocytes is common (white arrow in inset of (**c**)). Occasionally, dual infection of hemococcidan sporozoite (arrow in (**d**)) and *Hepatozoon* gamont (arrowhead in (**d**)) can be observed.

**Figure 4 microorganisms-11-02374-f004:**
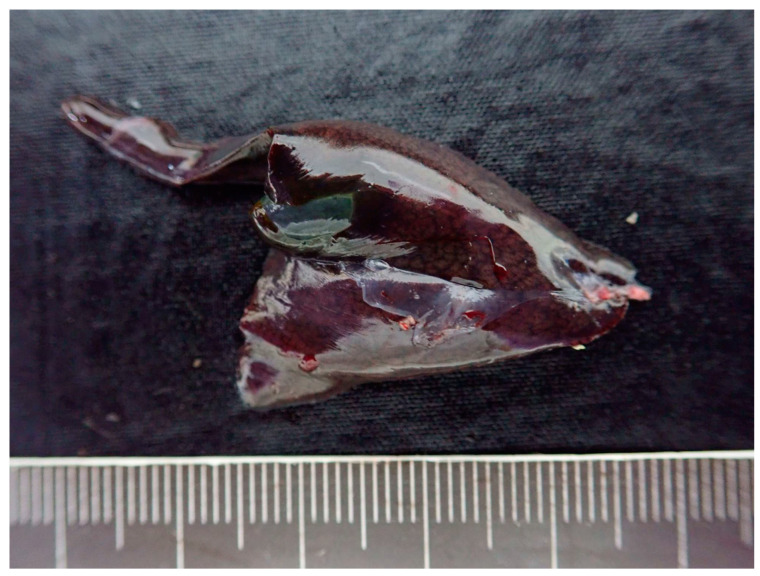
The liver of iguana 1 was diffusely dark green and slightly swollen.

**Figure 5 microorganisms-11-02374-f005:**
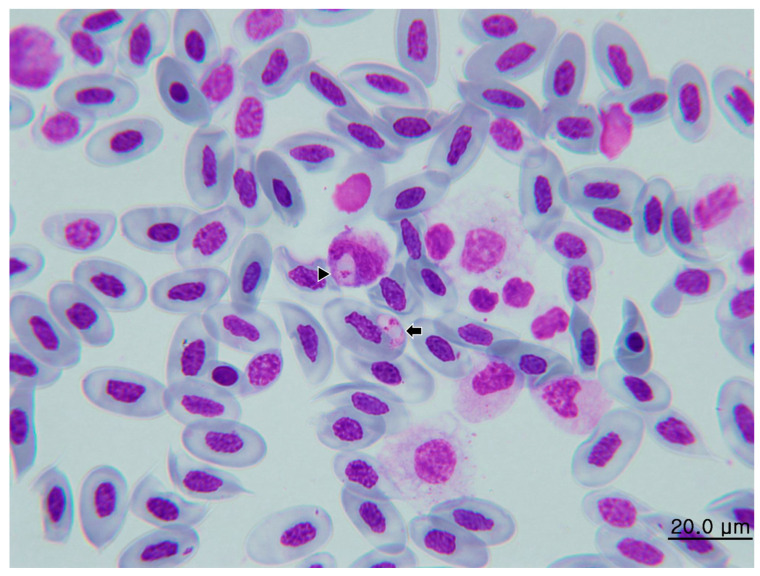
Sporozoites of *Lankesterella desseri* n. sp. can be observed in erythrocytes (arrow) and one monocyte (arrowhead) of iguana 1.

**Figure 6 microorganisms-11-02374-f006:**
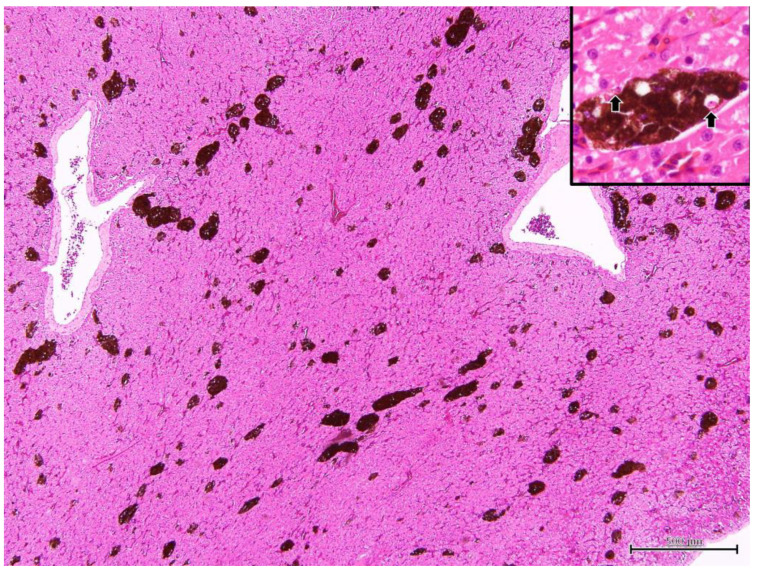
Under the section of the liver, the melanomacrophage centers (MMCs) are hyperplastic. Within the MMCs, numerous hemococcidan sporozoites can be observed in the cytoplasm of melanomacrophages (arrows in inset).

**Figure 7 microorganisms-11-02374-f007:**
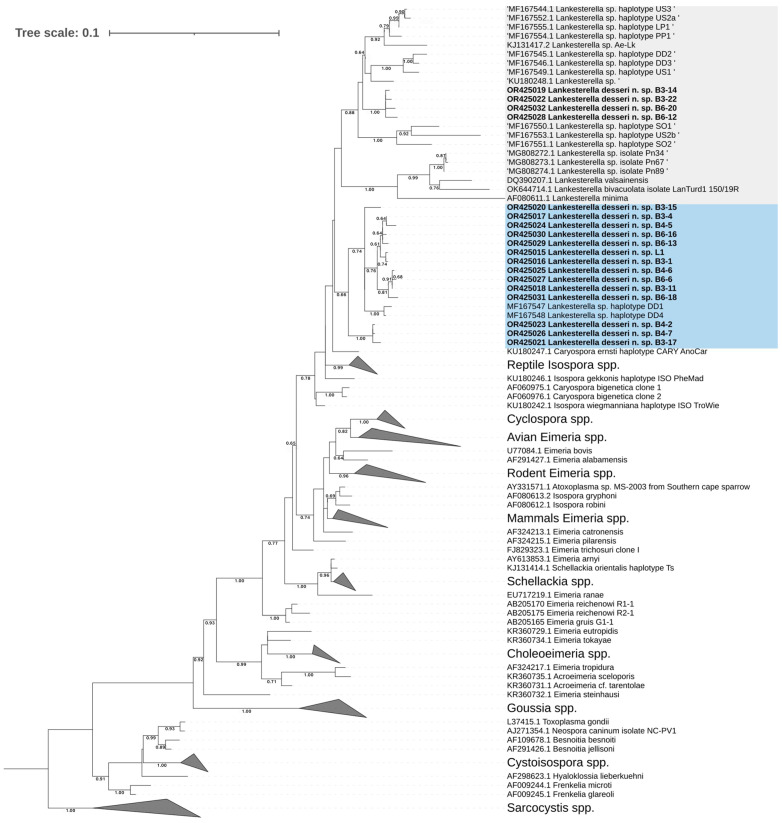
Maximum likelihood tree based on the partial 18S rDNA sequences of *Lankesterella* parasites and other related apicomplexans. Of the 18 *Lankesterella* sequences obtained in this study, 14 are clustered with haplotypes DD1 and DD4 (MF167547 and MF167548) obtained from the desert iguana *Dipsosaurus dorsalis* (blue shade). The other 4 haplotypes from iguanas 3 and 6 form a monophyletic clade and are closely related to other *Lankesterella* branches (gray shade). The sequences obtained in the current study are presented in bold font. GenBank accession numbers are provided for all the sequences. Bootstrap values > 60% (based on 1000 replicates) are shown at branch points.

**Figure 8 microorganisms-11-02374-f008:**
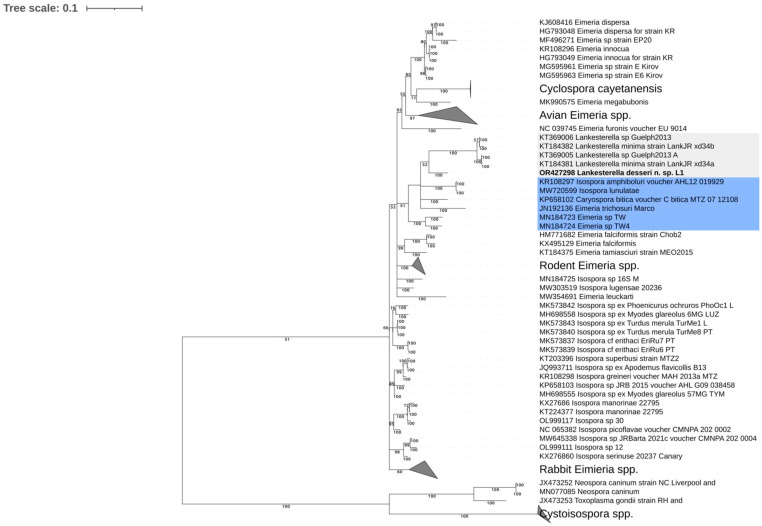
Bayesian inference tree based on the partial mitochondrial cytochrome c oxidase subunit I gene sequences of *Lankesterella* parasites and other related apicomplexans. The sequences obtained in the current study are presented in bold font. GenBank accession numbers are provided for all the sequences. Bayesian posterior probabilities > 50 are shown at branch points. The sequence obtained in this study forms a well-supported clade (posterior probability = 100) along with the closest sister group to the other 4 *Lankesterella* sequences (gray shade). The *Lankesterella* clade is grouped with several genera of the Eimeriorina (blue shade).

**Figure 9 microorganisms-11-02374-f009:**
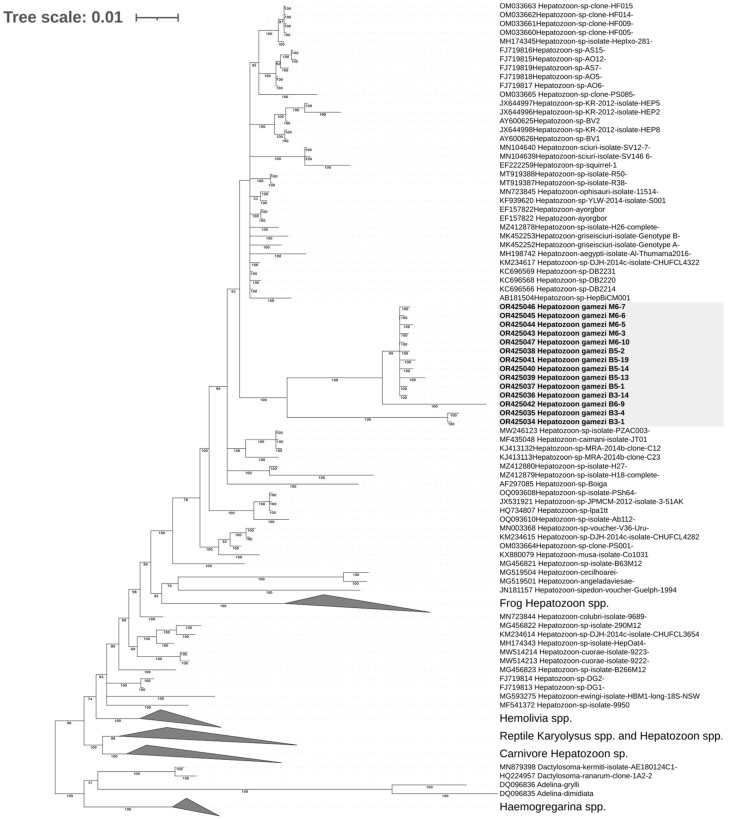
Bayesian inference tree based on the partial 18S rDNA sequences of *Hepatozoon* parasites and other related apicomplexans. The sequences obtained in the current study are presented in bold font. GenBank accession numbers are provided for all the sequences. Bayesian posterior probabilities > 50 are shown at branch points. All *Hepatozoon* sequences obtained in this study formed a new monophyletic clade with high statistical support (posterior probability = 100, gray shade) and were inserted into a large clade composed of *Hepatozoon* strains obtained from reptilians or rodents.

**Table 1 microorganisms-11-02374-t001:** Body condition, intensity of hemoparasites, and serum biochemistry of the 6 black spiny-tailed iguanas in this study.

Items or Status	1 ^a^	2 ^a^	3 ^a^	4 ^a^	5 ^a^	6 ^a^	Reference Interval ^b^
Weight (grams)	-	224.5	87.5	131.3	128.2	88.0	
Dehydration	10–15%	0–5%	5–10%	5–10%	5–10%	5–10%	
Hemococcidia intensity (/15,000 erythrocytes)	8	6	9	48	0	22	
*Hepatozoon* intensity (/15,000 erythrocytes)	0	2	18	11	10	3	
Aspartate aminotransferase	-	20	45	60	20	18	0–97 U/L
Bile acid	-	<35	<35	<35	35	<35	2.6–30.3 µmol/L
Creatine kinase	-	2854	2790	-	613	1421	174–8768 U/L
Uric acid	-	2.3	1	1.2	1.5	0.9	0–8.2 mg/dL
Glucose	-	157	182	374	258	221	169–288 mg/dL
Calcium	-	10.6	11.8	13.5	10.4	10.1	6–18 mg/dL
Phosphate	-	8.8	7.7	14.8	11.1	9.2	2.5–21 mg/dL
Total protein	-	7.2	8.5	9.7	7.3	8.1	4.1–7.4 g/dL
Albumin	-	3.3	3	3.8	3.2	3	2.1–2.8 g/dL
Globulin	-	3.9	5.5	6	4.1	5.1	2.5–4.3 g/dL
Potassium	-	3.8	2.6	4.9	5.1	3.7	1.3–3 mmol/L
Sodium	-	161	153	167	161	152	158–183 mmol/L

^a^ Serial number of black spiny-tailed iguanas. ^b^ Retrieved from *Exotic Animal Formulary, 5th Edition* [32].

**Table 2 microorganisms-11-02374-t002:** Morphometry of sporozoites of *Lankesterella desseri* n. sp. and gamonts of *Hepatozoon gamezi*. The values and length and width are presented in micrometers (μm).

Parasite Haplotype	Number	Length ^a^	Width ^a^
*Lankesterella desseri* n. sp.	59	7.3–15.9 (12.25 ± 1.62)	1.87–5.49 (3.71 ± 0.84)
*Hepatozoon gamezi* vacuolar gamonts	28	14.40–18.61 (16.55 ± 0.93)	5.65–9.45 (7.61 ± 0.90)
*Hepatozoon gamezi* basophilic gamonts	20	15.37–18.48 (17.00 ± 1.06)	5.60–9.95 (7.54 ± 1.79)

^a^ Minimum and maximum values are provided, followed in parentheses by the arithmetic mean and standard deviation.

## Data Availability

The data presented in this study are available in the Appendix A and in the GenBank database (https://www.ncbi.nlm.nih.gov/genbank/, accessed on 10 August 2023) (accession numbers OR425015–OR425032, OR425034–OR425047, and OR427298).

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
