# Peer review of "Reevaluation of Hemoparasites in the Black Spiny-Tailed Iguana (*Ctenosaura similis*) with the First Pathological and Molecular Characterizations of *Lankesterella desseri* n. sp. and Redescription of *Hepatozoon gamezi"

_microorganisms, 2023, doi:10.3390/microorganisms11102374_

Round 1

Reviewer 1 Report

Chang et al present an interesting morphological and molecular study on the hemococcidia of six captive black iguanas. They found for the first time Lankesterella in this species. They describe it as a new species based on the morphology and phylogenetic position. They also redescribe Hepatozoon gamezi, providing molecular and phylogenetic information for the first time. I have only a few comments prior to consider this study ready to be published.

 Title: spiny-tailed iguana

Line 19: further confirmed they belong

Lines 34 and 35: First, blood is a tissue. Second, the phrase in lines 34-36 needs to be rephrased.

 unicellular protists which have various life cycles between development in the tissues and blood of a range of vertebrate hosts and blood-35 feeding invertebrate vectors

Perhaps something like: …unicellular protists, which life cycles develop in several tissues of both vertebrate and blood-feeding hosts

Line 37:

“ in all tetrapod classes host” correct to “in hosts of all tetrapod classes”

Line 43: Correct to Lainsonia

Line 49: change “avian” to “birds”. Also, is there any urodele amphibian infected by Lankesterella, if not maybe change “amphibians” to “anurans” and lizards.

 Lines 54-55: A strong statement. I recommend a better search of the literature on reptile haemococcidians.

Line 57: correct to Adeleorina.

Line 64: Adeleorina

Line 70: “the Central America” change to “Central America”.

Line 84: “with the chief complaint of thermal burn” Not sure what the authors mean here. Please, rewrite.

Line 93: 25-gauge

Lines 93-95: any reference supporting this method of blood extraction?

Lines 118-199: what does mean “for routine histhopathological examination”? Please, be more specific for others to come be able to repeat the methods.

Lines 296-297: “The liver section of iguana 1 revealed no any positive signal against the anti-Toxo-296 plasma gondii antibody.” Better say: “The liver section of iguana 1 was negative against the anti-Toxo-296 plasma gondii antibody”.

Lines 309-311: COI PCR was positive for Lankesterella or Hepatozoon? The genbank code is not available yet at the NCBI website.

Line 320: “descent” support perhaps means “decent” support? Perhaps is better to say “relatively” strong support.

Lines 482-483: The two species, Ctenosaura similis and Dipsosauus dorsalis, do belong to the lizard family Iguanidae. Lankesterella seems to be a quite host specific genus of parasites, as it has been recently pointed out by Venkatachalam, A. K. S. B., Čepička, I., Hrazdilová, K., & Svobodová, M. (2023). Host specificity of passerine Lankesterella (Apicomplexa: Coccidia). European Journal of Protistology90, 126007. Similarly, there is significant co-phylogenetic/co-speciation signal between Schellackia parasites and their lacertid lizard hosts as shown by Megía-Palma, R., Martínez, J., Cuervo, J. J., Belliure, J., Jiménez-Robles, O., Gomes, V., ... & Merino, S. (2018). Molecular evidence for host–parasite co-speciation between lizards and Schellackia parasites. International journal for parasitology48(9-10), 709-718.

Therefore, I encourage the authors to comment on the possibility that the two Lankesterella detected in Dipsosaurus and Ctenosaura are closely related in the phylogeny because both infect the same lizard family.

Line 493: There is an interrupted phrase.

Lines 506-507: It is perhaps likely. However, it is speculative because only mites from this iguana was positive and following the methods described by the authors, the mites were not allowed to digest the blood meal prior to fixate them. Thus, the ingestion of infected blood from an infested iguana might per se favor the detection of Hepatozoon in the mite which does not necessarily mean that it is the vector.

None

Reviewer 2 Report

The authors address an important issue in their research, the methodology is described in detail, however, they do not mention that they also performed DNA extraction from mites found in an iguana.

Do the authors have information in which conditions the iguanas were kept before showing clinical signs? Were they isolated or were they in contact with other reptiles? If so, they could include it, I consider it important to know the origin of the infection.

Did you manage to identify the nematodes found in iguana 1? Could it be associated with the clinical signs and death of the iguana?

What happened to the iguanas after sampling? were they treated with any medication? Were they seized to prevent other reptiles from becoming infected with the parasites found?
